# Multifunctional SERS Chip for Biological Application Realized by Double Fano Resonance

**DOI:** 10.3390/nano14242036

**Published:** 2024-12-19

**Authors:** Weile Zhu, Huiyang Wang, Yuheng Wang, Shengde Liu, Jianglei Di, Liyun Zhong

**Affiliations:** 1Guangdong Provincial Key Laboratory of Photonics Information Technology, Guangdong University of Technology, Guangzhou 510006, China; 2112203214@mail2.gdut.edu.cn (W.Z.); jiangleidi@gdut.edu.cn (J.D.); zhongly@gdut.edu.cn (L.Z.); 2School of Optoelectronic Science and Engineering, South China Normal University, Guangzhou 510006, China; 2021010155@m.scnu.edu.cn (H.W.); wangyh919@foxmail.com (Y.W.); 3Guangdong Provincial Key Laboratory of Nanophotonic Functional Materials and Devices, South China Normal University, Guangzhou 510006, China

**Keywords:** Raman spectroscopy, SERS, dual Fano resonances, biological samples

## Abstract

The in situ and label-free detection of molecular information in biological cells has always been a challenging problem due to the weak Raman signal of biological molecules. The use of various resonance nanostructures has significantly advanced Surface-enhanced Raman spectroscopy (SERS) in signal enhancement in recent years. However, biological cells are often immersed in different formulations of culture medium with varying refractive indexes and are highly sensitive to the temperature of the microenvironment. This necessitates that SERS meets the requirements of refractive index insensitivity, low thermal damage, broadband enhancement, and other needs in addition to signal enhancement. Here, we propose a SERS chip with integrated dual Fano resonance and the corresponding analytical model. This model can be used to quickly lock the parameters and then analyze the performance of the dual resonance SERS chip. The simulation and experimental characterization results demonstrate that the integrated dual Fano resonances have the ability for independent broadband tuning. This capability enhances both the excitation and radiation processes of Raman signals simultaneously, ensuring that the resonance at the excitation wavelength is not affected by the culture medium (the refractive index) and reduces heat generation. Furthermore, the dual Fano resonance modes can synergize with each other to greatly enhance both the amplitude and enhanced range of the Raman signal, providing a stable, reliable, and comprehensive detection tool and strategy for fingerprint signal detection of bioactive samples.

## 1. Introduction

Noble metal nanostructures are commonly used to manipulate [1,2], confine [3], or enhance [4,5] the electromagnetic field. These mechanisms are facilitated by the strong coupling of the excitation and the collective resonance of conduction band electrons in the nanostructures, known as surface plasmon polaritons (SPPs) or localized surface plasmons (LSPs), which have been widely used in Surface-enhanced Raman spectroscopy (SERS). By synergistically combining the near-field interactions of the structures with a nanogap [6,7] and diffractive coupling of that with a relatively larger distance [8], the cascade amplification of the Raman signal can be achieved [9]. Since the SERS of pyridine molecules was first discovered on rough silver electrodes in 1974 [10], it has rapidly evolved into a biochemical detection tool capable of single-molecule detection [11,12]. Yet, such achievements, which benefit from the “hot spots” in the randomly self-assembled nanogap, suffer from poor spectral reproducibility and controllability, limiting their practical applications in biochemical samples lacking chemical bonds to couple with the nanogap.

Accordingly, advanced strategies leverage metal nanostructured surfaces [13], especially metal–insulator–metal (MIM) structures, to improve the reliability and controllability of the Raman signal enhancement. Since their optical properties depend on rich horizontal and vertical geometric degrees of freedom, they offer flexible design options for SERS chips [14,15,16,17]. Reproducible and tunable Raman enhancements have been established by first coupling the excitation light with the rationally designed resonator arrays in the upper metallic layer and then interacting with the near-field resonators in the lower metallic layer in the MIM structures [18,19,20]. One typical example is the double resonance SERS chips, which achieve significant Raman signal enhancements by simultaneously overlapping two resonances with the excitation light and the Raman Stokes wave [21,22,23,24]. In recent years, a new double resonance named Fano resonance, formed by coherent coupling between a broadband super-radiation mode generated from the resonator arrays and a narrowband sub-radiation mode from the smoothed metal film, can be used to create huge near-field electric field enhancement [25,26,27]. Although there is much research about Fano resonance [28,29,30], there are still challenges in fully utilizing it due to difficulties in simultaneously tuning its peak and dip to overlap with both excitation light and Raman Stokes waves [31].

Despite the critical role of large electric field enhancement in high-performance SERS chips, there are still some subtle but crucial issues for practical SERS applications, especially those for label-free living cells [32], including (1) photothermal and autofluorescence induced by the large electric field at the excitation light, which can damage the biological activity and affect the authenticity and reliability of the measurements; (2) the narrow spectral enhanced range of one single SERS chip, which poses a challenge in covering the wide distribution range of complicated biochemical components; and (3) resonance drift due to changes in the refractive index when changing different cultures, which will severely bias the results of the same cell under the same experimental conditions. Therefore, there is an urgent need for SERS chips with multifunctions beyond Raman signal enhancement. However, integrating such multifunctions into a single SERS chip presents challenges when considering contradictory conditions or limitations—such as balancing high Raman signal enhancement with low photothermal generation or achieving wideband enhancement tuning while maintaining fixed Raman excitation enhancement.

Here, we demonstrate how to integrate two Fano resonances with different mechanisms into one single SERS chip to achieve the multifunction of being less photothermal, having wide band enhancement coverage, and being cultural-solution insensitive. Our experimental findings reveal that the second Fano resonance plays a crucial role in Raman enhancement, which resonates at the Raman Stokes. Importantly, with the help of the second Fano resonance, the demand for enhancement of the first Fano resonance at excitation is reduced, thus greatly lowering the photothermal nature and autofluorescence of the chip. Furthermore, we observed that the first Fano resonance remained stable when changing cultural solutions with different refractive indices or tuning the second Fano resonance in the broadband Raman Stokes region. Additionally, we found that the tuning of the two Fano resonances is independent and can be achieved by adjusting only one parameter on a single chip. This multifunctional SERS chip opens up new possibilities for the comprehensive detection of bioactivators such as living cells in realistic environments and provides a flexible and universal platform for biomedical applications.

## 2. Methods

### 2.1. Sample Fabrication

The SERS chips investigated in this study were fabricated on ITO-coated glass using electron beam lithography (EBL). We utilized a spin coater to apply PMMA 950K (polymethyl methacrylate) with a concentration of 12% onto a clean ITO substrate surface at a speed of 4000 rpm for 90 s, ensuring an even distribution to form a smooth film layer of 120 nm on the ITO substrate. Next, we used the ultra-high-resolution capability of the electron beam lithography machine (PIONEER TWO) to precisely expose the designed nanohole regions on the PMMA film layer. Following that, we dissolved the exposed PMMA areas with a developer and terminated the development process with a fixer. The development time was 60 s and the fixation time was 30 s, resulting in a nanohole array that allows for precise control of the size. The nanohole array was then coated with a 50 nm thick gold film using electron beam evaporation. The morphologies of the fabricated SERS chips were examined using atomic force microscopy (NT-MDT). The preparation process is shown in Figure 1.

### 2.2. Optical Characterization

The reflection spectra of the SERS chip were collected by an Angle-Resolved Micro-Spectrometer (ARMS, ideaoptics, Shanghai, China), which was equipped with an enhanced charge-coupled device (970-BVF, Andor, Oxford, UK). The prepared SERS substrate was immersed in a p-mercaptobenzonitrile solution of 10 mL at a concentration of 40 μmol/L for 3 h, and then the Raman reporter molecules (p-mercaptobenzonitrile) adhered to the surface of the Au film through strong Au-S bonds to form a dense self-assembled monolayer. Excess molecules were then washed away with a large amount of alcohol. Raman scattering measurements were performed using a Raman spectrometer (Renishaw Invia, London, UK) with an excitation laser light of 632.8 nm.

### 2.3. Numerical Simulation

In this work, the reflection spectra and the electric field enhancements of the SERS chip were simulated using the finite element method (FEM)-based numerical simulations (COMSOL Multiphysics v5.6, https://www.comsol.com). The refractive indices of PMMA, SiO_2_, and Au were derived from experimental data [33] or taken from the previous literature [34].

## 3. Results and Discussion

The multifunctional SERS chip is designed as a three-layer film system, similar to the MIM system, as illustrated in Figure 2. It is fabricated simply by coating a single layer of gold film to the PMMA nanohole array with a depth *h* greater than the gold film thickness *t*. The top gold nanohole array and bottom gold nanodisk array are geometrically complementary to each other, supporting a narrowband sub-radiation mode based on SPP-Bloch waves and another based on Surface Lattice resonance (SLR), respectively. Moreover, the entire system supports a third continuous super-radiation mode with ultra-wideband enhancements. When the two sub-radiation modes are coherently coupled with the super-radiation mode at different wavelengths, it results in a special double Fano resonance: the first Fano resonance remains stationary at the excitation wavelength while changing the environmental refractive index and/or shifting the second Fano resonance in a broadband region due to their distinct mechanisms. In accordance with the two-step enhancement mechanism of SERS [31], the Raman signal enhancement factor EF is expressed as
(1)$$\mathrm{EF}(\omega_{0},\omega_{R})\;\approx M_{Exci}\cdot M_{Rad}\approx {\left|\frac{E_{loc}(\omega_{0})}{E_{0}(\omega_{0})}\right|}^{2}\cdot {\left|\frac{E_{loc}(\omega_{R})}{E_{0}(\omega_{R})}\right|}^{2}$$
where $M_{Exci}$ and $M_{Rad}$ is the excitation and radiation enhancement, respectively, and $E_{loc}(\omega_{0})$ is the local fields produced by a plane wave at the excitation light and $E_{loc}(\omega_{R})$ at the Raman Stokes light. Configuring such a second resonance overlapping the Raman Stokes as the radiation enhancement is expected to reduce the photothermal and fluorescence caused by the excitation enhancement compared to conventional methods at the same Raman signal-enhancement level.

To demonstrate the mechanisms of the double Fano resonance in the proposed SERS chip, we first examine a three-layer film system whose nanohole period *P* is significantly shorter than the excitation wavelength, as depicted in Figure 2. Based on effective medium theory (EMT) [35] and the metasurface-assisted law of refraction and reflection (MLRR) [36], thin films with subwavelength structures can be considered equivalent to smooth films with an isotropic equivalent refractive index which is determined by the material and the proportions that constitute this thin layer [37], as shown in Figure 3a. For SERS applications typically involving normal incidence, the equivalent refractive index can be described according to the EMT:(2)$$n_{\mathrm{eff}}={\left[(1-F)n_{1}^{2}+Fn_{2}^{2}\right]}^{1/2}$$
where $n_{1}$, $n_{2}$ represents the refractive index of different materials and F is the volume proportion of the material.

In such a scenario, the three-layer film system enables a continuous super-radiation mode similar to Fabry–Perot (F-P)-type resonance. This is achieved through the interference of multiple transmissions and reflections of incident light, resulting in a reflection coefficient expressed as follows (refer to the Appendix A for details):(3)$$r_{eff}=\frac{r_{k}+r_{k+1}e^{-j2\delta_{k}}}{1+r_{k}r_{k+1}e^{-j2\delta_{k}}}$$
where $r_{k}$ represents the equivalent reflectance coefficient of the layer *k* obtained from the Fresnel formula and $\delta_{k}$ represents the phase delay caused by layer *k*. From Equation (3), it is evident that the phase shift determines the overall reflectivity of the SERS substrate, which is related to the intensity of the Fabry–Perot (F-P)-type resonance. Once the equivalent refractive index is set, it is determined by the thickness of the three layers. To make the intensity of the Fabry–Perot (F-P)-type resonance comparable to the amplitudes of the SLR mode and/or the SPP modes, thereby increasing the coupling efficiency between them, and also considering the requirements of the coating process, we adopted the film thickness that results in a reflectance of approximately half (as shown in Figure 1), and its reflectance is illustrated in Figure 3b. As depicted in Figure 3b, the reflection curve of the super-radiation mod appears from visible to infrared according to Formula (2), which plays an important role in ultra-broadband mode in the following Fano resonance. Due to these parameters determining the F-P mode amplitudes, which are comparable to the amplitudes of the SLR mode or the SPP mode when coupled, it is evident from the reflection coefficient represented by the cyan curve in Figure 4f that the values of the two Fano dips are very small. This implies that more energy is confined to the surface of the SERS substrate without being reflected into the external environment, leading to a stronger near-field electromagnetic enhancement.

Previous research has demonstrated that periodic metal nanostructures can stimulate the SPP and generate narrowband SPP-Bloch resonance efficiently [20], while the nanodisk arrays can help to form SLR [13], whose resonance spectrum is much narrower than that of a single nanodisk. Accordingly, two different narrowband resonances and one broadband resonance are simultaneously supported in the proposed SERS chip, as schematically shown in Figure 3c. To investigate the interaction between them, we conducted finite element method (FEM) numerical simulations to analyze the reflection spectra. Initially, we set the parameters identical to those in Figure 2 but removed the gold nanodisk arrays to observe the interaction between the SPP-Bloch mode and Fabry–Perot (F-P) mode. The simulation results depicted in Figure 3d show the efficient excitation of surface charge distribution representing the SPP-Bloch wave due to the presence of periodic nanoholes. Furthermore, an asymmetric line shape reflection representing Fano resonance is observed (Figure 3f, red curve). By varying the nanohole period *p*, as shown in Figure 3e (the white dash line), it is evident that there is a linear shift in this asymmetric line shape, indicating coupling between F-P mode and SPP-Bloch mode. Subsequently, we introduced gold nanodisk arrays at the bottom layer, leading to another asymmetric line shape representing a second Fano resonance alongside with first one (Figure 3f, cyan curve). This observation suggests a coupling between F-P mode and SLR mode. Notably, there is minimal change in the location of the first Fano resonance with the appearance of the second one, indicating the independent tunability of the two Fano resonances. The independent formation mechanism of these Fano resonances significantly simplifies the SERS tuning complexity and allows for the flexible adjustment of resonance peaks without concern for mutual effects on each other’s resonances—a scenario typically requiring multiple parameter adjustments and re-evaluation using traditional methods. In other words, the excitation and the Stokes resonances can be independently and simultaneously tuned to double boost the SERS signals.

To further understand the mechanism behind the two Fano resonances, we developed an analytical model based on Couple Mode Theory (CMT) [38] to study the interaction inside the proposed SERS chip. CMT has proven to be a powerful tool for characterizing resonance coupling behaviors [39] and has been utilized to calculate the far-field spectrum including reflection, transmission, or scattering [40,41]. When incident light interacts with the proposed SERS chip supporting three different resonances, the proposed analytical reflection model can be expressed as follows (please refer to the Appendix A for details):(4)$$R={\left|r_{\mathrm{eff}\;}+\frac{A_{0}e^{i\varphi_{0}}}{i\left(\omega -\omega_{0}\right)+\gamma_{1}}+\frac{A_{R}e^{i\varphi_{R}}}{i\left(\omega -\omega_{R}\right)+\gamma_{R}}\right|}^{2}$$

Here, the first term $r_{\mathrm{eff}\;}$ is the reflectivity coefficient of F-P resonance described in (3), the second represents SPP-Bloch resonance, and the third is SLR. The parameters A, φ, ω, γ are the amplitude, phase, frequency of resonance, and the Lorentz line width expressing a system loss, and index 0 and R represent resonances at excitation and Raman Stokes light, respectively.

To examine the ability to describe the double Fano resonances, we investigated whether the full-field reflection spectrum simulated by the FEM can be reproduced by the proposed analytical model. As a practical example, when the 632.8 nm commercial laser is selected as the excitation, the period *p* can be determined to be ~315 nm according to the Fano dip shown in Figure 3d. Based on this information, we obtained the analytical model by fitting the parameters in Equation (4). To assess the model accuracy and generalization, we varied the nanohole diameter *d* while keeping the other parameters fixed, and the spectral map calculated using the analytical model and the simulated are shown in Figure 4a and Figure 4b, respectively. As comparatively observed, the analytical results align well with the simulated results across various wavelengths and periods. Of particular interest is the observation that both the analytical and simulated spectral maps indicate a consistent fixation of the first Fano resonance around 632.8 nm, while the second Fano resonance continuously redshifts from ~630 nm to ~780 nm as the nanohole diameter *d* changes. This wideband and continuous tunability of the second Fano resonance further confirms that our model has accurately captured the underlying physics, making it an ideal tool for designing the double Fano resonance SERS chips and predicting the performance, thereby reducing the reliance on heavy full-field numerical simulations and facilitating the rapid determination of basic parameters for specific SERS applications.

It is important to emphasize that although the two Fano resonances can be independently tuned to cover the excitation and Raman Stokes simultaneously, they are not mechanically pieced together to double boost the Raman signals. Indeed, Fano resonance is an important mechanism for enhancing the near-field electromagnetic spectrum, whose sharp dip usually indicates the characteristic wavelength of the enhanced electric field. However, the Fano dip does not necessarily overlap with the peak of the near-field enhanced spectrum [42], especially in metal structures with relatively high ohmic loss, which is determined by the so-called Fano shape parameter that determines the asymmetry degree of the spectrum [43]. To achieve a better SERS performance by making use of the proposed model, we decomposed the analytical spectra with a nanohole diameter *d* of 220 nm and 270 nm (the black dotted line and blue dotted line in Figure 4a) into three modes to investigate their interactions. The corresponding decomposed F-P mode (the red curve), SPP-Bloch mode (the cyan curve), and SLR mode (the yellow curve) are shown in Figure 4c and Figure 4d, respectively. With the assistance of the analytical model, the corresponding phase-wavelength curves can also be obtained and are shown in Figure 4e,f. The Fano resonance is observed in the reflection curves (the green curves) shown in Figure 4c,d, where it coincides with the appearance of the SPP-Bloch mode or the SLR. When the nanohole diameter *d* ≤ 220 nm, the amplitude of SLR resonance is significantly smaller than that of the SPP-Bloch resonance due to the insufficient excitation of nanodisk arrays, and its resonance wavelength nearly overlaps with that of the SPP-Bloch resonance. Consequently, only one Fano resonance is formed. As depicted in Figure 4e, the phase-wavelength curve for the SPP-Bloch resonance is sharp, while that for F-P is smooth. This unique phase relationship results in the simultaneous generation of two different modes–in phase and out of phase–when they undergo coherent coupling, thereby forming a typically asymmetric line shape for Fano resonance. When the nanohole diameter *d* > 220 nm, the SLR amplitude experiences significant growth compared to the SPP-Bloch resonance and redshifts from the excitation wavelength (Figure 4d, yellow curve), forming the second Fano resonance which boosts the Raman Stokes. Interestingly, as shown in the phase-wavelength curve in Figure 4f, both the SPP-Bloch resonance and SLR exhibit in-phase vibration with the Fabry–Perot (F-P) resonance on the long wavelength side of their resonances. This indicates that when they are coherently coupled with F-P resonance, they can generate a larger electric dipole together, thereby enabling more effective radiation of Raman signals into free space during far-field radiation. The emergence of this second Fano resonance not only cascades the enhancement of Raman signals but also compensates for the limited coverage of the first Fano resonance, thus facilitating enhanced Raman signals across a wideband range from near field to far field.

In Raman enhancement applications of biological cells, the cells are often immersed in various formulations of a culture medium. Changes in the refractive index of the culture medium can lead to a drift of resonance, resulting in a loss of enhancement for the excitation light. To investigate whether the proposed SERS chip can accommodate different scenarios involving culture solutions with varying refractive indexes, we simulated a situation where the background refractive index changes from 1.33 (water) to 1.5. From Figure 5a (the black dotted line), it is evident that the first Fano resonance used to enhance the excitation light remains nearly unchanged with variations in the refractive index, indicating that the SERS chip exhibits good refractive index insensitivity and can be effectively utilized in diverse application scenarios involving culture media. Although there is a slight drift in the second resonance, this drift can be effectively suppressed by appropriately adjusting the size of the nanohole diameter d based on our analysis above.

Due to the different formation mechanisms of these two Fano resonances, we can tune them without interfering with each other. Specifically, for Raman signal detection, while maintaining the first resonance unchanged to enhance the excitation light, the second resonance can be continuously and broadly tuned to enhance the Raman Stokes light, resulting in a double boost for Raman signals. As shown in Figure 5b, FEM simulation demonstrates that by varying the nanopore diameter d, the average electric field enhancement at excitation and Raman Stokes wavelengths above 2 nm of the SERS chip can be tuned. This result is consistent with the analytically predicted result (the chain line), indicating that the electric field enhancement near 632.8 nm remains constant regardless of changes in the nanohole diameter *d*, while the second resonance shifts from ~641 nm to ~781 nm. According to the Raman wave number, the tuning range reaches 3000 cm^−1^. Importantly, this tuning strategy has three advantages over traditional methods: (1) it is continuous rather than discrete; (2) only one parameter needs to be tuned without requiring adjustments to multiple parameters; (3) tuning does not require a change in the film thickness. These advantages greatly simplify tuning and fabrication processes by eliminating the need for re-coating layers with different thicknesses to optimize the SERS chip performance. Furthermore, fabrication only requires a single plating with 100% gold film utilization, reducing the costs associated with gold materials.

Based on the analysis and the simplified fabrication process mentioned above, we prepared the proposed SERS chip. Figure 6c shows the 3D topography of one of the SERS chips characterized by atomic force microscopy with the parameters of *h* = 120 nm, *t* = 50 nm, *p* = 315 nm, and *d* = 270 nm. For comparison, SERS chips with the same parameters but varying nanohole diameter *d* were prepared on the same substrate. The p-mercaptobenzonitrile molecule, which has a broadband Raman Stokes distribution range, was chosen as the Raman reporter. The whole SERS chip was submerged in the p-mercaptobenzonitrile solution and then rinsed with a large amount of alcohol to ensure consistent concentrations of Raman reporters coupled to SERS chips with different nanohole diameters. We characterized the optical response by reflectance spectroscopy when varying the nanohole diameter. As shown in Figure 6a, when *d* = 220 nm, it can be seen from the simulation and experimental results that there is only one valley, which, according to the proposed analytical model, is due to the resonance amplitude of the SLR being too small at this time (Figure 4c), and there is only a Fano resonance formed by coupling the F-P mode with the SPP-Bloch mode. This results in a relatively small Raman signal enhancement, as shown in the red curve in Figure 6d, and an enhancement compared to no structure (the gold film), as shown in the inset. When the nanohole diameter *d* is increased to 250 nm under the same excitation light power, the Raman signal is further enhanced due to the double Fano resonance that begins to establish, as shown in the cyan curve in Figure 6d. However, since the second Fano resonance is at ~670 nm and the corresponding Raman Stokes wavelength is ~900 cm^−1^, the Raman peak at about 2227 cm^−1^ still has a small enhancement, and the signal at about 2926 cm^−1^ is even more undetectable. When the nanohole diameter *d* continues to increase to 270 nm, the SLR mode is significantly enhanced, even exceeding the SPP-Bloch mode (Figure 4d), forming a significant double Fano resonance, as shown in Figure 6b; the second Fano resonance at ~700 nm not only enhances the Raman signal at 1584 cm^−1^ near the resonance by 39.7 times (compared to the single Fano resonance, the red curve in Figure 6d) but also amplifies the Raman signal at 2250 cm^−1^ by a factor of 28.9 by making use of the mechanism of building up a large dipole by the three-mode in-phase vibration of F-P, SPP-Bloch, and SLR (Figure 4f). More interestingly, thanks to such a broadband enhancement mechanism, we also observed a significantly enhanced Raman signal at 2926 cm^−1^. This phenomenon shows that the SLR mode can not only form a second Fano resonance with the F-P mode to achieve a direct enhancement effect but also synergize with the other two resonance modes supported in this SERS chip to play a good role in broadband enhancement.

## 4. Conclusions

In view of the special needs of biological samples, we propose a non-nanogap Raman-enhanced strategy based on double Fano resonance and the corresponding analytical model, through which the parameters to meet the needs of specific scenarios can be quickly locked and the performance of the dual resonance Raman enhanced chip can be analyzed. The effectiveness of the proposed analytical model and the function of SERS chip suitable biological samples are verified by a cross-comparison of theory analysis, simulation, and experiment. We integrated dual Fano resonance into a single SERS chip to reduce thermal damage to biological activity and the fluorescence background during Raman signal detection. Interestingly, whether the refractive index is changed by changing the culture medium or the second Fano resonance is tuned by changing the nanohole diameter, the first Fano resonance is not affected and drifts, and the excitation enhancement of the Raman signal can be stably achieved at the preset wavelength. Based on the stable enhancement brought about by the first Fano resonance, the SERS spectral was further enhanced by the second Fano resonance at the stage of Raman signal emission, which not only achieves the above multifunctions but also exerts the synergistic effect of multi-mode resonance, greatly enhancing the amplitude and detection range of Raman signals, providing a stable, reliable, and comprehensive detection tool and strategy for fingerprint signal detection of biologically active samples.

## Figures and Tables

**Figure 1 nanomaterials-14-02036-f001:**
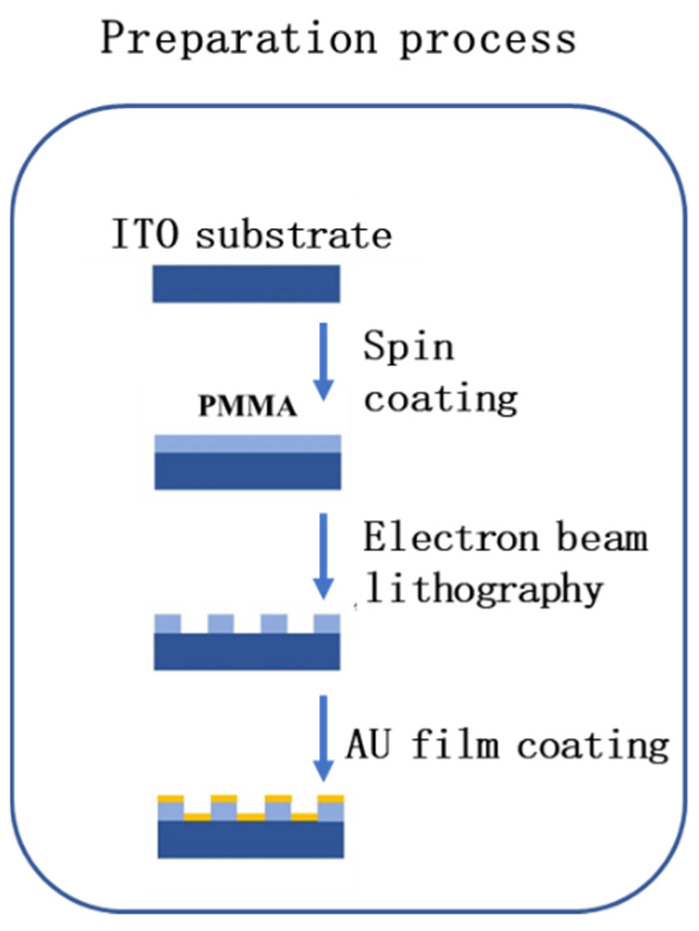
Schematic diagram of the SERS chip fabrication steps.

**Figure 2 nanomaterials-14-02036-f002:**
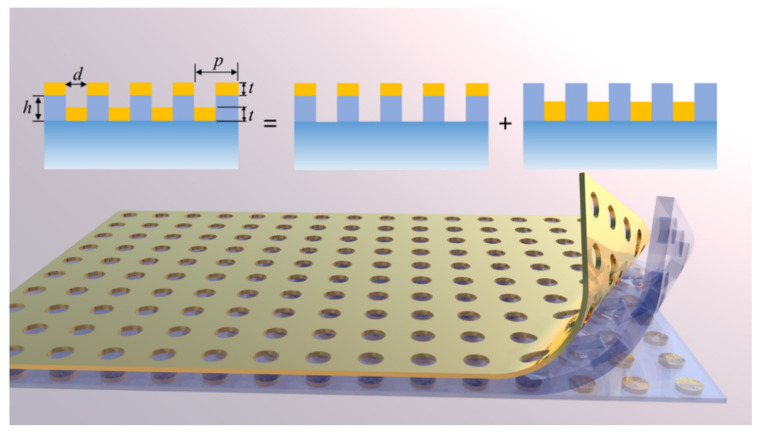
The schematic of the proposed SERS chip, which consists of a 3-layer structured thin film system constructed on a silica substrate. The thickness of the upper and lower layers containing gold *t* = 50 nm, the thickness of PMMA film *h* = 120 nm, the diameter of the nanohole is *d*, and the period *p* = 315 nm.

**Figure 3 nanomaterials-14-02036-f003:**
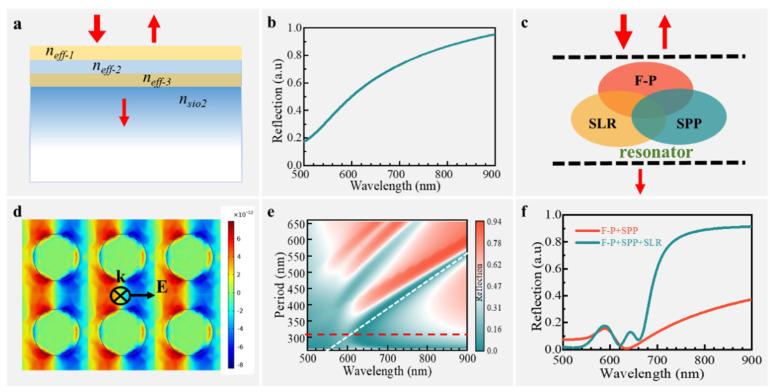
Formation mechanism of double Fano resonance in the proposed SERS chip. (**a**) The SERS chip with a subwavelength period is equivalent to a 3-layer thin film system under equivalent medium theory; (**b**) F-P resonance-like reflectance spectra in the equivalent 3-layer thin film system; (**c**) schematic diagram of the three resonance modes (F-P-like, slr, and SPP-Bloch) supported simultaneously in the SERS chip and their interactions; (**d**) the surface charge distribution of the SPP-Bloch resonance formation calculated by FEM simulation, where k represents the incident light wave vector and E represents the direction of electric field polarization; (**e**) reflectance spectra of SPP-Bloch mode with different nanopore periods coupling to the F-P mode, where the red curve represents the reflection spectrum with a nanohole period *p* = 315 nm; and (**f**) comparison of reflectance spectra in SERS chip when there is interaction between only two resonance modes (SPP-Bloch and F-P) and three resonance modes (SPP-Bloch, SLR, and F-P).

**Figure 4 nanomaterials-14-02036-f004:**
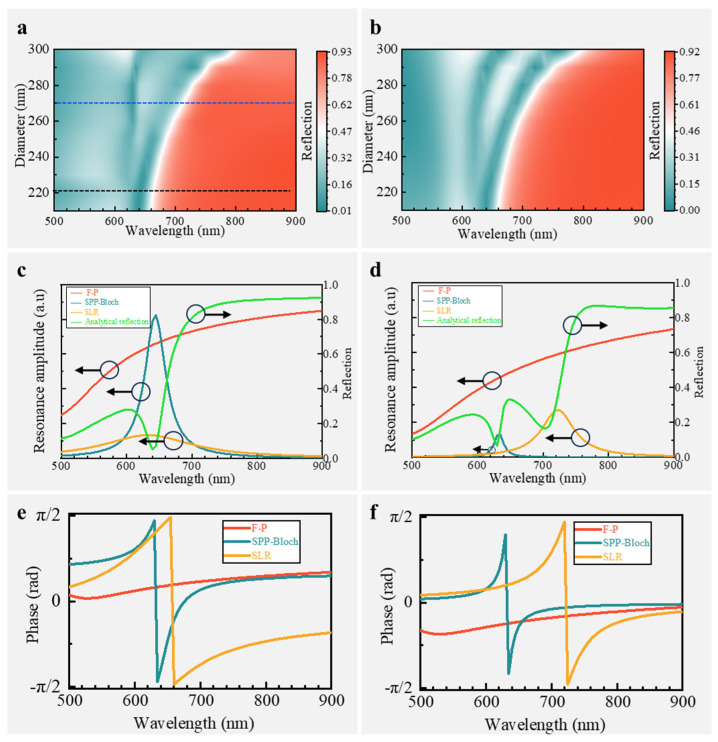
Analytical model analysis of the double Fano resonance mechanism in the SERS chip. (**a**) Reflection mapping using the proposed analytical model; (**b**) reflection mapping using FEM simulation; (**c**,**d**) reflection spectrum (right ordinate) of nanohole diameter d = 220 nm (black dot in (**a**)) and d = 270 nm (blue dot in (**a**)) and the corresponding decomposition of resonance modes (left ordinate), respectively; (**e**,**f**) phase relationship among the three resonances generating the double Fano resonance corresponding to (**c**,**d**), respectively.

**Figure 5 nanomaterials-14-02036-f005:**
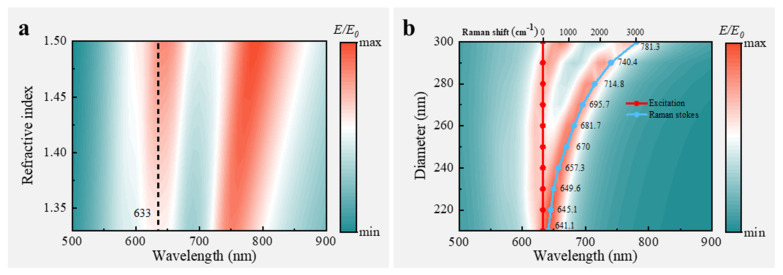
(**a**) Resonance spectrum with refractive index changing from 1.33 to 1.5; (**b**) electric field enhancement spectra calculated using FEM simulation and analytical model (dot curve) while tunning the second Fano resonance.

**Figure 6 nanomaterials-14-02036-f006:**
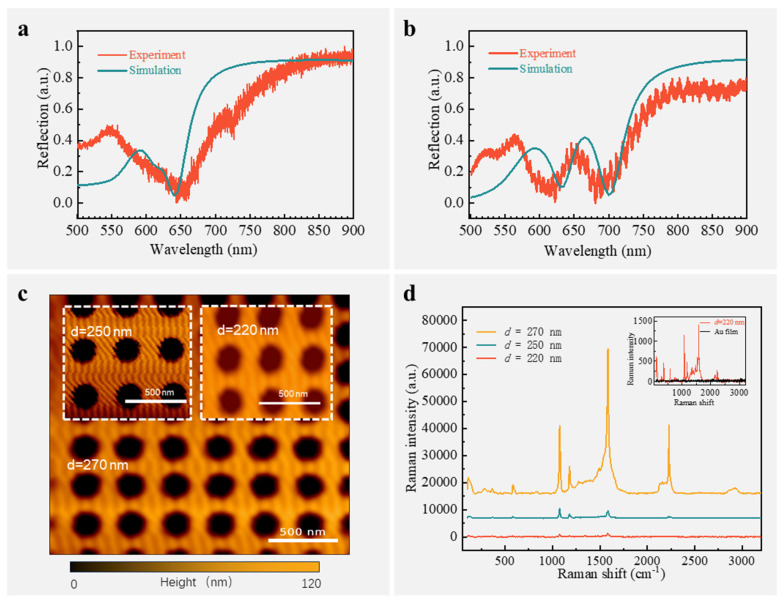
(**a**,**b**) Experimental and simulated reflection spectrum of SERS chip with a nanohole diameter *d* =220 nm and *d* = 270 nm, respectively; (**c**) three-dimensional topography of the SERS chip with nanohole diameter *d* = 220, 250, and 270 nm characterized by atomic force microscopy; (**d**) comparison of Raman signals enhanced by SERS chips with nanohole diameters *d* = 220, *d* = 250, and *d* = 270 nm, respectively, with the inset showing a comparison of Raman signals from a flat gold film and a SERS chip with a nanohole diameter of *d* = 220 nm.

## Data Availability

The data underlying the results presented in this paper are not publicly available at this time but may be obtained from the authors upon reasonable request.

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
