# Peer review of "Multifunctional SERS Chip for Biological Application Realized by Double Fano Resonance"

_nanomaterials, 2024, doi:10.3390/nano14242036_

Round 1
Reviewer 1 Report
Comments and Suggestions for Authors
The authors in this article propose and investigate by simulations and experiments a SERS device using the effect of dual Fano resonance for enhanching Raman signals form biological samples. This elps minimize thermal damage to biological samples and reduce fluorescence interference during Raman signal detection.
The article is well-structured and presented. And the theoretical investigation agrees very well with experimantal results.
The conclusions are supported by the experiments.
I recommend publication as is.
Author Response
Comment 1:
The authors in this article propose and investigate by simulations and experiments a SERS device using the effect of dual Fano resonance for enhanching Raman signals form biological samples. This elps minimize thermal damage to biological samples and reduce fluorescence interference during Raman signal detection.
The article is well-structured and presented. And the theoretical investigation agrees very well with experimantal results.
The conclusions are supported by the experiments.
I recommend publication as is.
Response 1:
We would like to express our gratitude to the reviewers for affirming our work.

Reviewer 2 Report
Comments and Suggestions for Authors
The present work reports on the design, modeling and experimental testing of novel type of SERS chips (substrates) based on the double-Fano resonance effect. Authors propose and fabricate a three layer periodic nanostructure composed by a PMMA film with vertical nanoholes with the top layer and the bottom of the holes being covered with gold. Authors show by means of theoretical simulations that this structure possesses three coupled plasmonic modes – two narrow-band, and one wide-band, which overlap spectrally and lead to a double Fano resonance.
The original idea behind this structure is to tune the two resonant frequencies to the laser frequency and to the Stokes frequency of the scattered radiation, respectively. As simulations show the resonant frequencies are insensitive to the refractive index of the environment, which significantly widens the functionality of this kind of a SERS chip. Raman experiments, reported by authors, demonstrate 20-40 times enhancement of the scattered intensity in comparison to a single-Fano resonance structures.
I find the work original, well motivated and supported both by theory and by experiments. I suggest accepting the manuscript for publication with only minor revisions:
1. It would be of interest to SERS practitioners if authors document in the text the absolute SERS enhancement factor from the fabricated structures in comparison to the bare analyte solution.
2. I would recommend a careful proofreading of the final manuscript for cleanup of the numerous typos in the text.
Author Response
Comment 1:
The present work reports on the design, modeling and experimental testing of novel type of SERS chips (substrates) based on the double-Fano resonance effect. Authors propose and fabricate a three layer periodic nanostructure composed by a PMMA film with vertical nanoholes with the top layer and the bottom of the holes being covered with gold. Authors show by means of theoretical simulations that this structure possesses three coupled plasmonic modes – two narrow-band, and one wide-band, which overlap spectrally and lead to a double Fano resonance.
The original idea behind this structure is to tune the two resonant frequencies to the laser frequency and to the Stokes frequency of the scattered radiation, respectively. As simulations show the resonant frequencies are insensitive to the refractive index of the environment, which significantly widens the functionality of this kind of a SERS chip. Raman experiments, reported by authors, demonstrate 20-40 times enhancement of the scattered intensity in comparison to a single-Fano resonance structures.
I find the work original, well motivated and supported both by theory and by experiments. I suggest accepting the manuscript for publication with only minor revisions:
It would be of interest to SERS practitioners if authors document in the text the absolute SERS enhancement factor from the fabricated structures in comparison to the bare analyte solution.
Response 1:
Thanks for reviewer’s valuable suggestions. When the size of the pump beam focal spot is kept constant, the Raman signal from the bare analyte solution involves the effective volume determined by the depth of field, as well as the number of Raman-reporting molecules within this effective volume. In SERS, the volume is determined by the ~100 nm region of electric field enhancement near the SERS substrate surface, and the number of Raman-reporting molecules is determined by the number of molecules adhering to the SERS substrate surface within this region. These factors differ from those in the case of bare analyte solution, which makes it difficult for us to distinguish specifically from which factor the enhanced Raman signal originates. To compare the Raman enhancement factor under the same conditions, we conducted Raman signal measurements on a smooth gold film, forming a dense monolayer similar to our SERS substrate on the gold film surface (please refer to the Optical characterization in methods section), which makes the number of Raman-reporting molecules and the effective enhancement volume consistent, with the results shown in the inset of Figure 5d.
Comment 2:
I would recommend a careful proofreading of the final manuscript for cleanup of the numerous typos in the text.
Response 2:
Following the reviewer's suggestion, we have carefully checked the manuscript for typos and have highlighted the corrections in red in the revised manuscript.

Reviewer 3 Report
Comments and Suggestions for Authors
This work presents the fabrication of a multifunctional SERS chip implemented with Fano resonance. Comparing experimental and simulated spectra greatly helps guide the development of unique SERS substrates. This type of work can provide interesting aspects for the analytical chemistry and materials science communities. However, a few important pieces of information listed below should be covered.
Some experimental procedures and interpretations of the results lack in detail throughout the manuscript.
For example, it would be helpful if the authors included a schematic diagram of the SERS chip fabrication steps in the “Sample fabrication” section. In addition, there was no clear explanation for the preparation of controlled diameters of nanoholes (200 nm – 270 nm).
Details of PMMA coating on the ITO substrate (used amount of PMMA to control 120 nm) and the details of Raman reporter treatment should be provided (used concentration of p-mercaptobenzonitirle, incubation time, etc.).
There were discrepancies between Figure 5 and the corresponding descriptions. Figure 5 presented the diameter range from 200-270 nm with no clear descriptions regarding 200 nm (is this part of below 220 nm related description?) and 230 nm (Figure 5d) in the manuscript. Similarly, the authors concluded that “When the nanohole diameter d continues to increase to 270 nm, the SLR mode is significantly enhanced, even exceeding the SPP-Bloch mode (Fig. 3d), forming a significant double Fano resonance, as shown in Fig. 5b, the second Fano resonance at ~700 nm not only enhances the Raman signal at 1584 cm−1 near the resonance by 39.7 times (compared to the single Fano resonance, red curve in Fig. 5d ), but also amplifies the Raman signal at 2250 cm−1 by a factor of 28.9 by making use of the mechanism of building up a large dipole by three-mode in-phase vibration of F-P, SPP-Bloch and SLR (Fig. 3f).” What about above 270 nm nanohole diameter?
Again, most of the results were explained based on the nanohole diameters of 220 nm and 270 nm. However, some results were presented using 230 nm (Figure 5d) (isn’t this 220 nm?). Clarification is needed.
It would be helpful if the authors could present all AFM topography images of the SERS chips with three different nanohole diameters.
In the introduction section, the authors may need to justify the selection of chip geometry they studied (e.g., selection of nanohole diameter, thickness of gold coating, thickness of PMMA film, etc..). These parameters can have a significant impact on the overall Fano resonance and refractive index.
The conclusion section should be rewritten based on their findings (how and which Fano resonance and refractive index directly affected the SERS spectra).
Author Response
Comment 1:
This work presents the fabrication of a multifunctional SERS chip implemented with Fano resonance. Comparing experimental and simulated spectra greatly helps guide the development of unique SERS substrates. This type of work can provide interesting aspects for the analytical chemistry and materials science communities. However, a few important pieces of information listed below should be covered.
Some experimental procedures and interpretations of the results lack in detail throughout the manuscript.
For example, it would be helpful if the authors included a schematic diagram of the SERS chip fabrication steps in the “Sample fabrication” section. In addition, there was no clear explanation for the preparation of controlled diameters of nanoholes (200 nm – 270 nm).
Details of PMMA coating on the ITO substrate (used amount of PMMA to control 120 nm) and the details of Raman reporter treatment should be provided (used concentration of p-mercaptobenzonitirle, incubation time, etc.).
Response 1:
Thank you for the reviewer's suggestions. We have included a schematic diagram of the SERS chip fabrication steps and additional content in the "Sample Fabrication" section.
The new content is as follows: "We utilized a spin coater to apply PMMA 950K (polymethyl methacrylate) onto a clean ITO substrate surface at a speed of 4000 rpm for 90 seconds, ensuring an even distribution to form a smooth film layer on the ITO substrate. Next, we used the ultra-high-resolution capability of the electron beam lithography machine (PIONEER TWO) to precisely expose the designed nano-hole regions on the PMMA film layer. Following that, we dissolved the exposed PMMA areas with developer and terminated the development process with a fixer. The development time was 60 seconds, and the fixation time was 30 seconds, resulting in an array that allows for precise control of the hole size". “The prepared SERS substrate was immersed in a p-mercaptobenzonitrile solution at a concentration of 40 μmol/L for 3 hours.” The additional content is displayed in red font in the revised manuscript.
Comment 2:
There were discrepancies between Figure 5 and the corresponding descriptions. Figure 5 presented the diameter range from 200-270 nm with no clear descriptions regarding 200 nm (is this part of below 220 nm related description?) and 230 nm (Figure 5d) in the manuscript. Similarly, the authors concluded that “When the nanohole diameter d continues to increase to 270 nm, the SLR mode is significantly enhanced, even exceeding the SPP-Bloch mode (Fig. 3d), forming a significant double Fano resonance, as shown in Fig. 5b, the second Fano resonance at ~700 nm not only enhances the Raman signal at 1584 cm−1 near the resonance by 39.7 times (compared to the single Fano resonance, red curve in Fig. 5d ), but also amplifies the Raman signal at 2250 cm−1 by a factor of 28.9 by making use of the mechanism of building up a large dipole by three-mode in-phase vibration of F-P, SPP-Bloch and SLR (Fig. 3f).” What about above 270 nm nanohole diameter?
Again, most of the results were explained based on the nanohole diameters of 220 nm and 270 nm. However, some results were presented using 230 nm (Figure 5d) (isn’t this 220 nm?). Clarification is needed.
It would be helpful if the authors could present all AFM topography images of the SERS chips with three different nanohole diameters.
Response 2:
We are sorry for the annotation errors in Figure 5d due to carelessness. We have corrected the values in Figure 5d from d=200 and d=230nm to d=220 and d=250nm, respectively, to eliminate the confusion caused by the inconsistency between the text and the figures.
Regarding the situation where the size of the nanohole is greater than 270nm, due to the resolution of the electron beam exposure machine we used, we were unable to successfully prepare SERS chips with a nanohole diameter greater than 270nm. However, according to the simulation results [Figure 3b], when the nanohole diameter is greater than 290nm, although the second Fano resonance can still be tuned towards longer wavelengths, the first Fano resonance also begins to redshift, which can deteriorate the Raman enhancement effect. Therefore, due to the limits of fabrication techniques and the requirement that the enhancement under commercial excitation light (632.8nm) must not drift, the size of the nanohole should not exceed 290nm.
Following the suggestion, we have presented the AFM topography images for all three nanohole sizes of the SERS chips in Figure 5c of the revised manuscript.
Comment 3:
In the introduction section, the authors may need to justify the selection of chip geometry they studied (e.g., selection of nanohole diameter, thickness of gold coating, thickness of PMMA film, etc..). These parameters can have a significant impact on the overall Fano resonance and refractive index.
Response 3:
Thanks for the reviewer’s suggestions. To avoid the introduction section from becoming bloated with excessive details, we have added the relevant content in the Results and Discussion section and discussed it in conjunction with Figure 2. The additional content is as follows:
"From equation (3), it is evident that the phase shift determines the overall reflectivity of the SERS substrate, which is related to the intensity of the Fabry-Perot (F-P) type resonance. Once the equivalent refractive index is set, is determined by the thickness of the three layers. To make the intensity of the Fabry-Perot (F-P) type resonance comparable to the amplitudes of the SLR mode and/or the SPP modes, thereby increasing the coupling efficiency between them, and also considering the requirements of the coating process, We adopted the film thickness that results in a reflectance of approximately half (as shown in Figure 1), and its reflectance is illustrated in Figure 2b ".
"Due to these parameters determining the F-P mode amplitudes, which are comparable to the amplitudes of the SLR mode or the SPP mode when coupled, it is evident from the reflection coefficient represented by the cyan curve in Figure 5f that the values of the two Fano dips are very small. This implies that more energy is confined to the surface of the SERS substrate without being reflected into the external environment, leading to a stronger near-field electromagnetic enhancement".
These additions are highlighted in red font within the revised manuscript.
Comment 4:
The conclusion section should be rewritten based on their findings (how and which Fano resonance and refractive index directly affected the SERS spectra).
Response 4:
Following the reviewer's suggestion, we have revised some of the conclusion content to highlight the role of the first Fano resonance in stably enhancing the excitation when changing the refractive index, as well as the emission enhancement role of the second Fano resonance in the Raman signal emission process. The revised parts are displayed in red font in the revised manuscript.

Round 2
Reviewer 3 Report
Comments and Suggestions for Authors
It appears that the authors have nicely revised the manuscript for publication.
One minor suggestion is to include the exact concentration of PMMA solution for spin coating and the volume of p-mercaptobenzonitrile solution prior to Raman analysis.
Newly added Figure 1 could be combined with Figure 2.
Author Response
Comments of reviewer3 Round 2
One minor suggestion is to include the exact concentration of PMMA solution for spin coating and the volume of p-mercaptobenzonitrile solution prior to Raman analysis.
Response:
According to the reviewer's suggestion, we have added the concentration of PMMA (12%) and the volume of p-mercaptobenzonitrile solution(10 ml), which were shown in red font in the revised manuscript.